# Artificial Intelligence in Pediatric Cardiology: A Scoping Review

**DOI:** 10.3390/jcm11237072

**Published:** 2022-11-29

**Authors:** Yashendra Sethi, Neil Patel, Nirja Kaka, Ami Desai, Oroshay Kaiwan, Mili Sheth, Rupal Sharma, Helen Huang, Hitesh Chopra, Mayeen Uddin Khandaker, Maha M. A. Lashin, Zuhal Y. Hamd, Talha Bin Emran

**Affiliations:** 1PearResearch, Dehradun 248001, India; 2Department of Medicine, Government Doon Medical College, Dehradun 248001, India; 3Department of Medicine, GMERS Medical College, Himmatnagar 383001, India; 4Department of Medicine, SMIMER Medical College, Surat 395010, India; 5Department of Medicine, Northeast Ohio Medical University, Rootstown, OH 44272, USA; 6Department of Medicine, GMERS Gandhinagar, Gandhinagar 382012, India; 7Department of Medicine, Government Medical College, Nagpur 440003, India; 8Faculty of Medicine and Health Science, Royal College of Surgeons in Ireland, D02 YN77 Dublin, Ireland; 9Chitkara College of Pharmacy, Chitkara University, Rajpura 140401, India; 10Centre for Applied Physics and Radiation Technologies, School of Engineering and Technology, Sunway University, Bandar Sunway 47500, Malaysia; 11Department of Biomedical Engineering, College of Engineering, Princess Nourah bint Abdulrahman University, P.O. 84428, Riyadh 11671, Saudi Arabia; 12Department of Radiological Sciences, College of Health and Rehabilitation Sciences, Princess Nourah bint Abdulrahman University, P.O. 84428, Riyadh 11671, Saudi Arabia; 13Department of Pharmacy, BGC Trust University Bangladesh, Chittagong 4381, Bangladesh; 14Department of Pharmacy, Faculty of Allied Health Sciences, Daffodil International University, Dhaka 1207, Bangladesh

**Keywords:** artificial intelligence, pediatric cardiology, pediatric cardiac surgery, machine learning, congenital heart diseases

## Abstract

The evolution of AI and data science has aided in mechanizing several aspects of medical care requiring critical thinking: diagnosis, risk stratification, and management, thus mitigating the burden of physicians and reducing the likelihood of human error. AI modalities have expanded feet to the specialty of pediatric cardiology as well. We conducted a scoping review searching the Scopus, Embase, and PubMed databases covering the recent literature between 2002–2022. We found that the use of neural networks and machine learning has significantly improved the diagnostic value of cardiac magnetic resonance imaging, echocardiograms, computer tomography scans, and electrocardiographs, thus augmenting the clinicians’ diagnostic accuracy of pediatric heart diseases. The use of AI-based prediction algorithms in pediatric cardiac surgeries improves postoperative outcomes and prognosis to a great extent. Risk stratification and the prediction of treatment outcomes are feasible using the key clinical findings of each CHD with appropriate computational algorithms. Notably, AI can revolutionize prenatal prediction as well as the diagnosis of CHD using the EMR (electronic medical records) data on maternal risk factors. The use of AI in the diagnostics, risk stratification, and management of CHD in the near future is a promising possibility with current advancements in machine learning and neural networks. However, the challenges posed by the dearth of appropriate algorithms and their nascent nature, limited physician training, fear of over-mechanization, and apprehension of missing the ‘human touch’ limit the acceptability. Still, AI proposes to aid the clinician tomorrow with precision cardiology, paving a way for extremely efficient human-error-free health care.

## 1. Introduction

The discipline of pediatric cardiology has evolved as a specialty over the past 60 years, deriving its roots from attempts to treat congenital heart diseases [1]. Congenital heart diseases (CHD), in the wake of the burden of mortality and morbidity they bring in, have always been an issue of concern. A total of 3.12 million babies in the United States were born with congenital heart disease, and around 13.3 million individuals are living with congenital heart anomalies [2]. CHD has a multifactorial etiology that consists of environmental stressors and genetic factors and accounts for 80% of all forms of CHD [3]. Pediatric heart diseases remain a global burden on health services and are associated with lifelong comorbidities that carry into adulthood, ultimately decreasing the quality of life for children [4]. The last decade has seen the increasing prevalence of hypertension and other ‘adult’ heart diseases among the pediatric population, especially adolescents, which has counterpoised the falling burden of rheumatic heart disease in the same age group.

Advancements in pediatric cardiology care and surgical technology have helped with significant reductions in mortality [5,6]. However, while high-income countries saw the mortality rates of CHD reduce by half, the needs for surgical care and advanced imaging for pediatric patients are unmet in middle- and lower-income countries [7]. The absence of a timely diagnosis for suspected pediatric heart diseases significantly delays timely treatment and often presents a diagnostic dilemma. The clinically suspected diagnosis of CHD is confirmed by echocardiography, which can even be conducted during pre- and postnatal screening [8,9]. The diagnosis thus requires adequate clinical suspicion, health infrastructure, and a skilled workforce [10].

The growth of artificial intelligence (AI) in the context of medicine has contributed immensely to the streamlining of clinical processes and decision making in health care since 1960 [11,12]. The concept of machine learning (ML) is integral to the evolution of AI. ML is defined as the ability of machines to learn tasks from a large amount of previous data and be able to predict the same for future instances [12]. As a result, AI has multiple key applications in the diagnosis, surveillance, prevention, and intervention of congenital heart diseases and has created major advancements in pediatric cardiology as a specialty [13]. Due to its widespread ability to improve the diagnostic value of cardiac magnetic resonance imaging (MRI), echocardiograms (ECHO), computer tomography (CT) scans, and electrocardiographs (ECG), AI can augment the diagnostic accuracy of pediatric heart diseases [14,15].

While AI has already found its applications in a multitude of specialties, the notion of its use in almost any medical specialty is conceivable. CHD is an interesting area where AI can be applied owing to the burden of CHDs in pediatric and adult populations. AI-based algorithms have expanded their application in various domains of pediatric cardiology including but not limited to screening, clinical examination, diagnosis, image processing, prognosis, risk stratification, and precision medicine [12]. Surprisingly, the full extent of AI applications in all stages of care for patients with congenital heart diseases has not yet been discussed. Evolving literature has pinpointed the efficiency of machine-learning algorithms in the interpretation of heart murmurs, a common sign of congenital heart diseases [16,17,18]. The recent utilization of deep-learning computer networks has demonstrated the ability to perform MRI segmentation, allowing clinicians to detect valvular defects simultaneously in all four of the heart’s chambers [19,20].

Based on our knowledge, there is only one systematic analysis of a case series of atrial septal defect repair with robotic assistance and AI; the full extent of AI applications in all stages of care for patients with CHD has not yet been discussed. The objective of this systematic review is to compile all existing literature on AI applications in the specialty of pediatric cardiology with a focus on CHD. The review will attempt to appraise the evolving literature and compile clinically relevant data that can serve as a source of health information for clinicians.

## 2. Materials and Methods

We conducted a scoping review in line with the PRISMA (Preferred Reporting Items for Systematic Reviews and Meta-Analyses) guidelines.

### 2.1. Data Sources and Searches

A literature search was conducted on PubMed, Scopus, and Embase by two authors (YS, NK) based on the following search terms: (“Artificial intelligence” OR “machine learning” OR “Deep learning”) AND (“Pediatric cardiology” OR “congenital heart disease”). The search was also refined using the MeSH Major Topic term “artificial intelligence”.

### 2.2. Article Selection

Duplicate studies were eliminated after the initial search, and two reviewers (NK, YS) independently evaluated the title and abstract to see if the articles qualified for a full-text review. An adjudicator (NP) overcame any disagreements. Both reviewers further evaluated the entire article in accordance with the inclusion and exclusion criteria for each potentially eligible study. A comprehensive review was ensured by screening the references to include additional studies (Figure 1).

### 2.3. Inclusion and Exclusion Criteria

The initial search was narrowed down by limiting the search to the English language and excluding animal studies. We limited our search to studies published between January 2002 and March 2022 because of the rise in interest in AI in the field of pediatric cardiology over the last 20 years. The inclusion criteria included observational studies (case–control, case series, and cross-sectional studies), experimental studies (randomized control trials), reviews, and expert opinions on the application of AI in pediatric cardiology. Studies were excluded if they lacked direct relevance to AI or were aimed at a population other than the pediatric (18 years) age group. Studies with no full text available, conference abstracts, papers, and book chapters were also excluded.

## 3. Results

The studies describing the role of AI in pediatric cardiology are compiled in Table 1, while Table 2 describes AI algorithms in pediatric cardiology.

### 3.1. AI and Heart Murmurs

AI has the potential to improve the validity of auscultatory findings for diagnosing CHD [87,88,89]. The limitations of objective performance data have restricted wide acceptability so far [90,91,92]. The auscultatory findings of CHD are integral to the clinical diagnosis and have the benefit of being a low-cost tool, but are subject to clinical expertise, which becomes a limitation in resource-limited countries and creates a need for support to the clinician [93,94,95,96] that is objective and reportable by even peripheral health workers. The lack of trained cardiologists at the peripheral level leads to an unavoidable miss of a timely clinical diagnosis of CHD, contributing to delayed intervention and thus a poor prognosis [97]. The emergence of AI-based digital stethoscopes and cloud reporting for telemedicine has helped with the timely diagnosis and early intervention of CHD in reported samples [68,70]. The AI-based technologies also seem affordable, helping the current interest. The use of an intelligent diagnostic system based on AI algorithms such as wavelet analysis and artificial neural networks (Figure 2) has shown a specificity of 70.5% and a sensitivity of 64.7% [71,72]. The developments in AI for detecting cardiac murmurs have shown promise in terms of sensitivity but still require clinical validation before wide clinical recommendation [73,98].

### 3.2. Image Processing with AI

#### 3.2.1. Chest X-ray

CHDs present with some classical chest x-ray signs that can help with suspicion of the disease, including boot-shaped heart (Tetralogy of Fallot), egg-on-string (Transposition of Great Arteries), snowman sign (Total Anomalous Pulmonary Venous Return), scimitar sign (Partial Anomalous Pulmonary Venous Return), gooseneck sign (Endocardial cushion defects), a figure of three (Coarctation of Aorta) and box-shaped hear (Ebstein anomaly) [99]. Chest radiography being easy, cost-effective, and readily available allows a direct or collaborative diagnostic approach to CHD. The evolution of deep learning models and machine learning models and the establishment of defined pediatric datasets have allowed the entry of AI for use in the pediatric population [74]. AI can be immensely beneficial in various steps of imaging such as ordering tests, reporting communication, enhancing the quality of images, and aiding radiologists in interpreting images [77].

#### 3.2.2. MRI

Cardiac MRI has evolved as a precise method for structural and functional evaluations of the heart [100]. The technology has evolved over the years from traditional techniques such as cardiac gating and the suspension of breathing to newer advanced techniques of high-field-strength magnets, high-performance gradient hardware and ultrafast pulse sequences. In the discipline of magnetic resonance imaging, the advancement of AI has allowed for shorter scan times, resulting in higher patient satisfaction; it also reduces errors by minimizing motion artefacts caused by patient movement. The segmentation of cardiac chambers helps visualize them better and aids in diagnosis. Currently, we do this manually, and a shift to AI will help with a faster diagnosis and reduce variation between different analysts [69].

#### 3.2.3. Echocardiography

New AI-based technologies have revolutionized modern medicine in obtaining fetal echocardiograms with improved precision and accuracy. Combining it with machine learning has been quintessential in making predictions of future variables associated with disease progression using retrospective patient data. It has helped in eliminating limitations associated with a lack of expertise in fetal echocardiology, fetal movements, and fetal heart size [19,74]. Machine learning-based systems have been useful in differentiating pathological versus physiological hypertrophic remodeling of the heart [59]. Deep learning models have demonstrated superiority in terms of sensitivity and specificity up to 76% and 88%, respectively, in diagnosing atrial septal defect (ASD) compared with pediatric cardiologists who demonstrated sensitivity and specificity of 53% ± 0.04 and 67% ± 0.10 respectively [35]. Automation, AI, and machine learning are game-changing as complementary tools to physicians and in areas with limited expert medical personnel or cardiologists [60]. Fetal intelligent navigation echocardiography (FINE) has presented as a novel method for fetal echocardiography using “intelligent navigation” technology to obtain nine standard views—four-chamber, aortic arch, three vessels, and trachea—and display abnormalities with great sensitivity helping to detect CHD [23,79].

#### 3.2.4. ECG

The use of deep learning (DL), an application of AI, in the field of adult cardiology has been well studied [64]. Its application in pediatric cardiology has become especially relevant due to its potential for allowing early diagnosis, and thus better prognosis, in congenital heart diseases (CHDs). DL models such as convolutional neural network (CNN) and recurrent neural network (RNN) are used in conjunction with electrocardiograms, which remain the staple diagnostic tool for CHDs, to provide enhanced diagnostic information that is otherwise only deduced with input from specialists [35]. Convolutional neural networks can perform image processing and classification, providing an advantage of extracting additional ECG information that would otherwise either be undetected [82]. While the efficiency and utility of using AI models with ECGs for CHDs is evident, the enhancement of the interpretability and training of deep learning models is still needed for widespread implementation [75].

### 3.3. Prognosis and Risk Stratification

AI-based algorithms have proven to be of great help for pediatric cardiology in the clinical examination, diagnosis, procedural planning, and management of cardiac interventions [12]. AI models have also aided with the extraction of patient data for risk stratification and ambulatory health monitoring from wearables [15]. Machine learning (ML)-based models such as optimal classification trees (OCTs) have accurately predicted mortality, postoperative mechanical ventilatory support time (MVST), and hospital length of stay (LOS) even with nonlinear data in patients with a history of congenital heart surgery [25]. Similarly, ML algorithm-based models, extreme gradient boosting (XGBoost), and RCRnet have accurately predicted preoperative mortality odds in patients with CHDs and statistically significant prognostic indicators along with risk stratification markers in patients with Tetralogy of Fallot and left-to-right shunt CHDs, respectively [41,53]. Importantly, the far-reaching utility of AI-based models in risk stratification and prognostic predictions is the trainability of models to work with various data cohorts [76].

### 3.4. Planning and Management of Cardiac Interventions

The current approach to planning and the execution of interventions in CHD relies primarily on generic treatment protocols derived from biological data and set guidelines. As such, there is a lack of tailor-made interventions based on each patient, which would drastically improve outcomes and post-intervention prognosis [78]. Artificial intelligence (AI), with its applications in the form of machine learning (ML) and deep learning (DL) among many, has emerged as a tool aiding the creation of personalized intervention plans as well as an accurate data extractor to ascertain potential post-operative sequelae [80]. The supervised machine learning models such as k-nearest neighbor classifier (KNN) and support vector machine classifier (SVM) have also aided in the intra-operative assessment of cardiac fluid kinematics, which are significant determinants of successful congenital heart surgeries [81]. The subsequent post-operative management of CHDs is imperative for maintaining the structural fluid dynamics and thus the prognosis. This could be done seamlessly with the help of recurrent neural networks based on ML models and deep learning models, allowing for an integrated system that can enhance survival outcomes in pediatric patients [101].

### 3.5. AI in Cardiac Surgeries

AI can revolutionize pediatric surgery in all stages of surgery: preoperative, intraoperative and post-operative. Preoperative risk assessment and decision-making can be made easier by the tremendous processing and analyzing capacity of AI-enabled algorithms. Surgical decision support systems using ANN can predict post-op outcomes and can prevent morbidity and mortality arising from poor risk assessment pre-operatively.

Tech-enhanced (Hitech) operation theaters can enable intraoperative interventions and decision support [83,102] and bring about a paradigm shift in telesurgery; it is especially useful in cardiac surgeries, where manual segmentation of retrieved images (CT/MRI) can take an unreasonable amount of time. Humans embarked on this path successfully when Xiaowei Xu et al. utilized AI to perform a cardiac surgery remotely in a patient having complications of long-standing ASD through the backbone of 5G technology. This was possible since AI replaced manual segmentation and provided accurate results in just two minutes compared to the traditional manual segmentation which takes 2–3 h even by experts [69]. In this situation, the authors helped a patient who could not be transported to another hospital due to her frail condition; we can extend the same advantage to a remote inaccessible location especially in rural areas and developing countries [103].

Postoperative AI can help in various aspects such as ambulatory patient monitoring post-discharge with AI wearables and automated risk stratification of patients to enable stricter follow-ups. Mahayni AA et al. described an ECG based AI algorithm that predicts ventricular dysfunction post-surgery, predicts long-term mortality in cardiac surgeries. Such algorithms can be developed and implemented for pediatric surgical procedures and drastically improve surgical outcomes [104].

### 3.6. AI in other Pediatric Heart Diseases

Kawasaki disease (KD) is an acquired pediatric vasculitis that can lead to coronary artery aneurysms and acute coronary syndrome [105]. Very little is known about the pathogenesis of Kawasaki disease but a recent study using AI-guided signatures reveals a shared pathology with Multisystem inflammatory syndrome in children (MIS-C), COVID-19 associated vasculitis in children. Both these syndromes share systemic inflammatory storm with similar cytokines such as IL15/IL15RA. Such AI-based investigative approaches will further elucidate its complex pathogenesis and help decipher novel diagnostic and therapeutic targets [86].

The appropriate and timely management of Kawasaki disease can significantly reduce the coronary complications associated with Kawasaki disease which contribute greatly to mortality in adulthood [106]. AI-based approaches can prove to be extremely crucial in predicting the risk of developing a coronary aneurysm. It is widely known that Kawasaki patients with intravenous immunoglobulin resistance are at a higher risk of developing coronary artery aneurysms, but most scoring models present for predicting the resistance are impractical. Wang T et al. describe a machine learning-based model on patient data that can predict intravenous immunoglobulin resistance in Kawasaki disease patients successfully. The scope for several such prediction models is possible with the advent of AI and can support clinical decisions to improve patient outcomes [84].

Distensibility changes in a coronary artery can be used to predict acute coronary disease in these patients. Benovoy M et al. calculated these changes in KD using an automated deep learning approach and correlated it with the severity of OCT (Optical coherence tomography) findings of KD-related CA damage [36]. Despite, the limited studies using AI-based approaches, AI has the potential to revolutionize risk stratification and prognosis estimation of Kawasaki disease as well as paving way for newer drug targets by elucidating the pathogenesis.

RHD—The screening of RHD requires the clinician’s expertise, and hence the grass-root level is limited by a lack of skilled human resources. Automatic diagnosis of echo-detected RHD is feasible and can form the core of the screening programs of the future covering the workload of experts [85]. Recent data has also shown promise for a convolutional neural network-based deep learning algorithm to identify heart sounds as ‘rheumatic’ with an overall accuracy of 96.1% having 94.0% sensitivity and 98.1% specificity [67]. Therefore, AI can form the backbone of future global screening programs for RHD.

### 3.7. AI Algorithms in Pediatric Cardiology

In many areas of pediatric cardiology, AI-based algorithms can be beneficial, including: (1) clinical examination and diagnosis, (2) image processing, (3) fetal cardiology, (4) prognosis and risk stratification, (5) precision cardiology, and (6) planning and management of cardiac interventions. Machine learning algorithms are a very promising tool for diagnosing and assessing critical and non-critical CHD; however, extensive research is still required to develop interpretable, robust, and generalizable models for clinical application, especially in light of the extreme heterogeneity of complex CHD. Various such AI algorithm models have gained interest over the past decade, and the present time has seen many motivating developments. The most relevant ones for our discussion are summarized in Table 2 and Figure 3.

## 4. Discussion

AI is rapidly growing, and its significance in clinical practice cannot be negated. It plays a crucial part in augmenting and standardizing care by adding to a physician’s skills and expertise. The role of AI in pediatric cardiology has greatly evolved over the past two decades (Figure 4). It picks up subtle or unrecognizable features, preventing missed diagnoses and leading to a better prognosis. Physicians can combine their clinical expertise with AI to enhance their outputs in the domains of prevention, predictive intervention, and health maintenance. AI can make use of continuous data received from wearable devices to offer insight into patients’ behaviors and health trends. These features of AI empower the professional to provide the best care to the patient earlier in the course of the disease, helping to improve prognosis and leading to better outcomes.

The current AI-based applications have greatly enhanced the diagnostics and prognostic areas of pediatric cardiology. The applications with the most promising potential are:(1)AI prediction algorithms: AI prediction algorithms can help assess patients’ risk based on left ventricular ejection fraction and predict post-surgery mortality outcomes based on predefined criteria.(2)Wearables: Wearables and mobile monitoring devices can help with ambulatory monitoring and early diagnosis. They can also help in educating patients about lifestyle modifications and health promotion.(3)EMR: Real-time analysis and the clustering of patients through EMR can help formulate research questions and aid the applicability of precision medicine.(4)Electrocardiography: ECG processing and classification based on deep learning-based algorithms can aid diagnosis.(5)Echocardiography: Deep learning-based programs such as FINE can also help process echo images to enhance the precision of detection of abnormalities.(6)Auscultation: AI algorithms such as wavelet analysis and ANN with digital stethoscopes promise to improve the accuracy of detecting abnormal heart sounds (murmurs).

AI seems to be the tool of rich and economically advanced countries, but it can be the magic wand for risk stratification troubles in lower- and middle-income countries (LMICs). AI has the potential to reduce global health inequalities [107]. It can help serve LMICs by bridging gaps in equity, availability, affordability, and accessibility [108]. For instance, the major addition of AI is a reduction in the costs of dimensions requiring expensive equipment and specialized expertise, including tools of screening and planning that are unavailable in most hospitals, especially in LMICs. AI-based innovations can help overcome resource-constrained environments.

The advent of AI has allowed smart tools to assist peripheral health workers in helping with a rough screening at the doorstep and in peripheral areas having poor doctor–patient ratios [60]. The availability of skilled cardiologists to diagnose CHD in remote locations will require a giant leap in these countries. AI-based medical equipment can aid peripheral health care providers in helping with screening and referral. It can also help in terms of teleconsultation and remote radiology [109]. Furthermore, with the availability of AI-based chatbots or virtual avatars and characters, we can offer greater help to populations suffering from stigmatizing pathologies (such as HIV/AIDS and psychiatric pathologies). This therefore can aid in improving access to health care and follow-up services. One of the biggest challenges to health care is language, with improving translation tools, AI can help cover this area well too [110].

### 4.1. AI: An Efficient Physician Assistant

The full-blown implementation of AI still is resisted by physicians and health care staff fearing inaccuracies, the exclusion of social variables, and possible unemployment. The mechanization of health care, loss of empathy, and human touch to this extent often invite skepticism from the traditional medical community. Despite the fear-mongering and skepticism of loss of human touch and jobs, we opine AI can be an efficient aid to physicians rather than a replacement [111]. It can streamline insurance reviews, provide real-time data analysis, and assist in research. It can reduce the burden on physicians and burnout rates in the medical community. Thus, application in appropriate areas can contribute to the overall enhancement of health care delivery and experience [13].

### 4.2. Challenges to AI in Pediatric Cardiology

The incorporation of AI into pediatric cardiology has several challenges. The data available for pediatric cardiology is still very limited which is necessary to train AI algorithms to identify, assess and reduce inherent biases and overfitting. Furthermore, the heterogeneity in cardiac anatomy and the rarity of individual disease entities make data accessibility and AI incorporation into pediatric cardiology difficult. We can cater to this limitation by pooling data from all the different hospitals to get a large data set [19]. Imaging in the pediatric population has another challenge owing to their smaller size and frequent movements during imaging, which lead to higher motion artifacts. This poses a technical challenge, for instance, requiring higher spatial resolution in the MRI [69].

The doctor and patient may hesitate to use AI to replace the current protocols. The incorporation of AI will require health care providers to learn interpret data and accurately understand the many model parameters or model architectures. This challenge is being tackled in different ways. For example, some works have incorporated more intuitive interfaces in the models aiding easier interpretation [75].

Additionally, AI is evolving and several ethical concerns are arising. These concerns include informed consent to data access, data security and privacy, algorithm fairness and biases, and transparency [112]. Currently, there are no well-defined laws or regulations in place to address the legal and ethical issues that may arise with AI use in health care settings. However, as AI continues to grow and be used, laws and regulations can certainly be explored to ensure algorithmic transparency and protect data privacy.

## 5. Conclusions

AI can be touted as the next revolution in medicine. From making physicians’ lives easier to enabling research with ease, it has variegated applications in the working environment as well as in the management of patients. Pediatric cardiology is a specialty requiring a great skill set in cognition and interpretation which makes it an ideal candidate for AI incorporation. AI has been successfully integrated into clinical examination, image interpretation, diagnoses, prognosis, risk stratification, precision medicine, and treatment in pediatric cardiology. The advent of AI has facilitated medicine to be more accurate and precise, but it is still a work in progress with challenges and limitations. Despite the roadblocks, we optimistically opine that AI with its current pace will streamline approaches in pediatric cardiology.

## Figures and Tables

**Figure 1 jcm-11-07072-f001:**
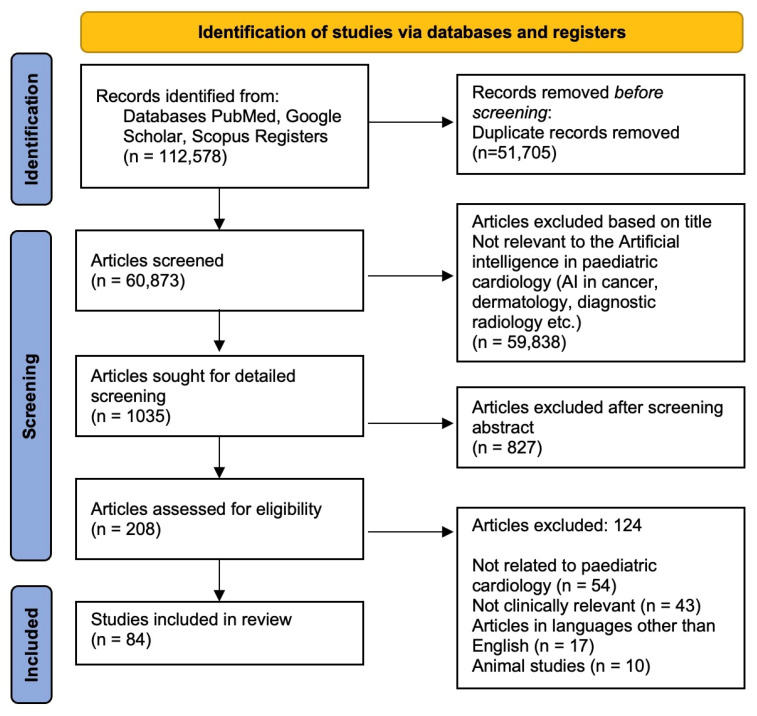
Flow diagram showing selection process of studies included.

**Figure 2 jcm-11-07072-f002:**
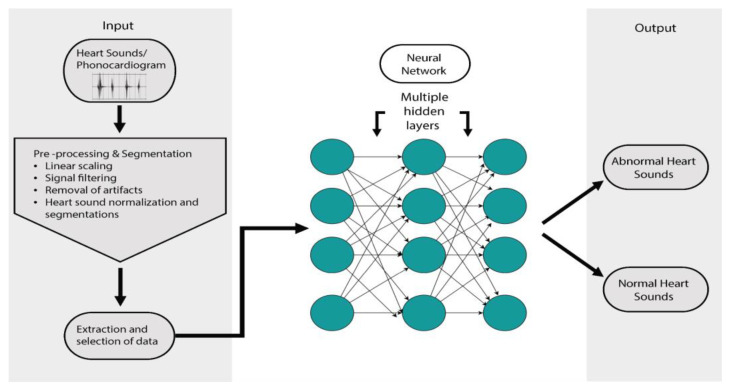
A model of application of neural networks in pediatric cardiology.

**Figure 3 jcm-11-07072-f003:**
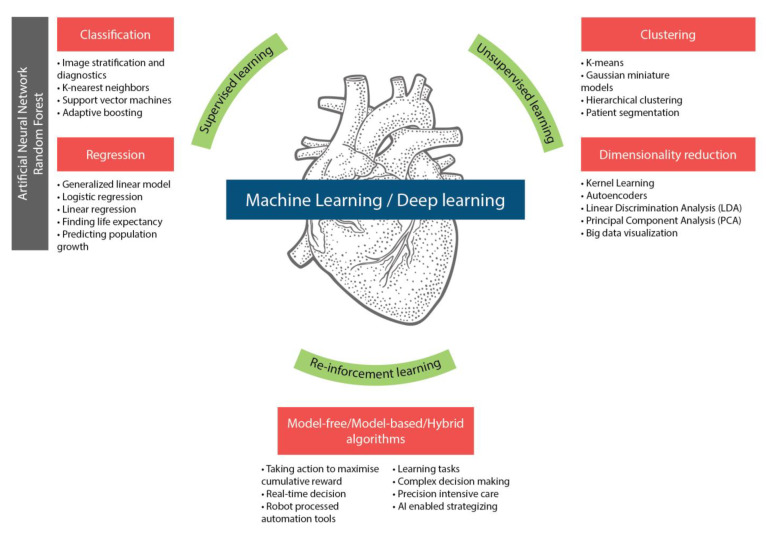
AI-based algorithms and pediatric cardiology.

**Figure 4 jcm-11-07072-f004:**
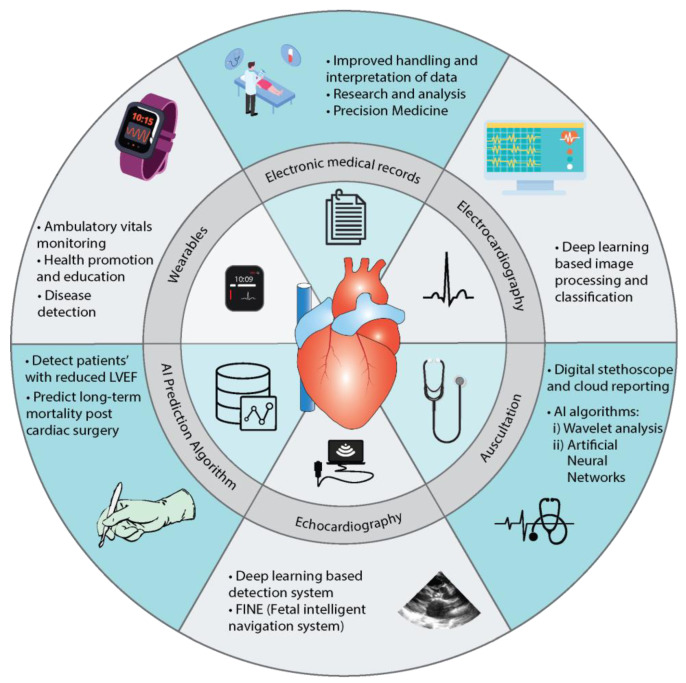
Diagnostic and prognostic applications of AI in Pediatric Cardiology.

**Table 1 jcm-11-07072-t001:** Pediatric Heart diseases and the role of AI.

Serial No.	Authors	Study Design	References	Pediatric Heart Diseases Covered	Applications of AI		
					AI in Diagnosis/Fetal Imaging	AI in Prognosis/Risk Stratification	AI in Cardiac Intervention
1	Jef Van den Eynde et al.	Review	[12]	General description	Clinical examination and diagnosis; image processing	Cardiovascular intervention planning and management; prognosis and risk classification.	Omics and precision medicine; fetal cardiology
2	Jingjing Lv et al.	Observational study	[21]	CHD	AI–AA platform revealed similar results to the experts’ face-to-face auscultation and reported high auscultation accuracy in detecting aberrant heart sounds.	-	-
3	Rhodri Davies et al.	Editorial	[22]	General description	Minor discernible fluctuations when ejection fraction was evaluated by a cardiac MRI expert were 8.7%, owing primarily to poor repeatability. Deep learning enables more accurate and precise analysis with quantifiable levels of confidence in the outcomes.	-	-
4	Sharib Gaffar et al.	Review	[15]	General description	-	With the aid of precise predictive risk calculators, ongoing health monitoring from wearables, and precision medicine, AI can assist in providing the best possible patient care.	-
5	Yeo et al.	Observational study	[23]	General description	FINE is an intelligent navigation technique that automatically acquires several anatomical views of the fetal heart during echocardiography to identify anomalies therein. In four cases, the instrument was able to show fetal heart structural malformations.	-	-
6	Arnaout et al.	Observational study	[24]	General description	Using 685 echocardiograms of fetuses between 18 and 24 weeks of gestation, supervised fully convolutional DL was used to (1) identify the 5 most crucial views of the fetal heart; (2) segment and measure the cardiac structures; and (3) differentiate between normal hearts, tetralogy of Fallot, and hypoplastic left heart syndrome.	-	-
7	Dimitris Bertsimas et al.	Observational study	[25]	CHD	-	For patients who underwent congenital heart surgery, machine learning (ML) models can predict mortality, postoperative mechanical ventilatory support time (MVST), and hospital length of stay (congenital heart surgery).	-
8	Ulrich Bodenhofer et al.	Editorial	[25]	CHD	-	In comparison with existing risk scores based on logistic regression on pre-selected factors, advanced machine learning is more accurate at predicting the results of valve surgery treatments. This strategy enables training models for the cohorts of certain institutions and is generalizable to other elective high-risk procedures.	-
9	Shaine A. Morris et al.	Expert opinion	[26]	CHD	Congenital illness, the most prevalent and fatal birth defect, could be more accurately diagnosed during pregnancy thanks to recent developments in machine learning.	-	-
10	Siti Nurmaini et al.	Observational study	[20]	CHD	Studies based on 1149 fetal heart images to predict 24 objects, including 3 congenital heart defect instances, 17 heart-chamber objects in each view, and 4 conventional fetal heart view shapes showed that the suggested model worked satisfactorily for segmenting standard views, with an intersection over union of 79.97% and a Dice coefficient similarity of 89.70%. Automatic segmentation and detection methods could significantly increase the number of CHD diagnoses.	-	-
11	Ai Dozen et al.	Observational study	[27]	VSD	To calibrate the output of U-net, cropping-segmentation-calibration (CSC) uses the time-series information of videos and specific section information. The mean intersections over union (mIoU) of 0.0224, 0.1519, and 0.5543, respectively, were used to assess the segmentation outcomes of DeepLab v3+, U-net, and CSC.	-	-
12	Makoto Nishimori et al.	Scientific report	[28]	Accessory pathway and WPW syndrome	A multimodal deep learning model based on 1D-CNN using ECG waveforms supported with CXR showed great accuracy in identifying AP location.	-	-
13	Tao Wang et al.	Observational study	[29]	General description	The adversarial learning mechanism focusing on the overall spatial structure and context consistency of myocardium showed more accuracy than the conventional method.	-	-
14	Yichen Ding et al.	Observational study	[30]	General description	The complete 3-D imaging of cardiac architecture and mechanics is made possible using light-sheet fluorescence microscopy. This innovative approach offers a solid foundation for post-light-sheet image processing and supports data-driven machine learning for the automated measurement of cardiac ultra-structure.	-	-
15	C Decourt et al.	Observational study	[31]	General description	The identification of the left ventricle in pediatric MRI using a generative adversarial network (GAN) segmentation approach was useful for the automatic analysis of cardiac MRI and for carrying out large-scale investigations based on MRI reading with a limited amount of training data.	-	-
16	Aapo L. Aro et al.	Editorial	[32]	ECG	Based on a single 12-lead electrocardiogram, AI may identify structural heart problems (AI-ECG).	-	-
17	W. Reid Thompson et al.	Observational study	[33]	CHD	An objective evaluation of an AI-based murmur detection algorithm showed promising results with a Sensitivity of 93% (CI 90–95%), specificity of 81% (CI 75–85%), with accuracy 88% (CI 85–91%) for the detection of pathologic cases. They also suggested that it could be used to compare the efficacy of other algorithms on the same particular dataset.	-	-
18	Saeed Karimi-Bidhendi et al.	Observational study	[34]	CHD	A GAN was devised that could accurately to synthetically augment the training dataset via generating synthetic CMR images and their corresponding chamber segmentations successfully.	-	-
19	Hiroki Mori et al.	Observational study	[35]		Using a deep learning model comprising a CNN and LTSMs, the researchers identified that the AI algorithm could identify the disease accurately with more sensitivity and specificity than pediatric cardiologists using electrocardiograms.	-	-
20	Benovoy M et al.	Observational study	[36]	Kawasaki disease		The degree of optical coherence tomography (OCT) observations of KD-related CA damage correlates with the degree of distensibility changes in the coronary artery (CA) of Kawasaki disease (KD) patients. When observed longitudinally, this reduced distensibility peaks at 1 year in KD patients and is more severe in those with persisting CA aneurysms.	-
21	Sweatt et al.	Observational study	[37]	Pulmonary arterial hypertension	Patients are categorized using machine learning (consensus clustering) into proteomic immune groups (cytokines, chemokines, and factors using multiplex immunoassay).	-	Different PAH immunological phenotypes with varying clinical risks are identified by blood cytokine patterns. These characteristics may help with mechanistic research on the pathobiology of disease and offer a framework for analysing patient responses to newly developed immunotherapy treatments.
22	Diller et al.	Observational study	[38]	CHD (transposition of great arteries—after atrial switch procedure or congenitally corrected TGA).	Use of deep machine learning algorithms trained on routine echocardiographic to detect the diagnosis.	-	Using machine learning algorithms that have been trained on common echocardiographic datasets, it is possible to determine the underlying cause of complex CHD and to perform a continuous, automated evaluation of ventricular function.
23	Li et al.	Observational study	[39]	CHD	-	To find the predictors that were substantially linked with CHD, ANN models such as univariate logistic regression studies and the traditional feed-forward back-propagation neural network (BPNN) model were used. Additionally, BPNN can be utilized to forecast a person’s risk of CHD.	-
24	Liu et al.	Observational study	[7]	CHD	-	An RCRnet model can preliminarily identify specific types of left-to-right shunt CHD and improve screening detection rate.	-
25	Tandon et al.	Observational study	[40]	CHD (rTOF)	-	-	The new mostly structurally normal (MSN) algorithm + rTOF algorithm showed improvements in LV epicardial and RV endocardial contours
26	Samad et al.	Observational study	[41]	CHD (rTOF)	-	Regression analysis previously failed to recognize the value of baseline variables, but machine learning pipeline did. Predictive models could help organise early interventions in high-risk individuals.	-
27	Diller et al.	Observational study	[42]	CHD	Deep learning (DL) algorithms enhance the de-noising of transthoracic echocardiographic images and removing acoustic shadowing artefacts.	-	-
28	Montalt-Tordera et al.	Observational study	[43]	CHD	Deep learning can improve contrast in LD cardiovascular magnetic resonance angiography (MRA) without sacrificing clinical utility.	-	-
29	Junior et al.	Observational study	[44]	CHD	-	Random forest (0.902) (a statistical model to ascertain mortality risk) gave top performing area under the curve and gave predictive variables that represented 67.8% of importance for the risk of mortality in the random forest algorithm.	-
30	Siontis et al.	Observational study	[45]	Hypertrophic cardiomyopathy	A deep-learning AI model can accurately identify juvenile HCM using a typical 12 lead ECG.	-	-
31	Tan et al.	Observational study	[46]	CHD	It is anticipated that a novel convolution neural network-based classification algorithm for CHD will be used in machine-assisted auscultation because it has increased heart sound classification accuracy, specificity, and robustness.	-	-
32	Baris Bozkurt et al.	Observational study	[47]	CHD	For automatic structural heart abnormality risk detection from digital phonocardiogram (PCG) signals, sub-band envelopes are preferred to the most often utilized features, and period synchronous windowing is preferred over asynchronous windowing.	-	-
33	Shaan Khurshid et al.	Observational study	[48]	General description	Estimates of the left ventricle’s mass-produced by deep learning from 12-lead ECGs and associated with incident cardiovascular disease.	-	-
34	Sabine Ernst et al.	Observational study	[49]	Intra-atrial baffle anatomy	-	-	SVTs might be safely and effectively eliminated using remote-controlled catheter ablation by magnetic navigation employing a retrograde strategy and precise 3D image integration.
35	Thomas Ernest Perry et al.	Observational study	[49]	General description	To effectively and efficiently utilize the potential of textual predictors, the Laplacian eigenmap technique embeds textual predictors into a low-dimensional Euclidean space.	-	-
36	Nikolaos Papoutsidakis et al.	Observational study	[50]	Inherited Cardiomyopathies	In order to effectively keep providers informed about pathogenicity assessments for any previously found genetic variant, the Machine-Assisted Genotype Update System (MAGUS) method of accessing ClinVar without specification to any specific gene or variant is proposed.	-	-
37	Shu-Hui Yao et al.	Observational study	[51]	PDA	-	-	When therapeutic drug monitoring is unavailable, the nine-parameter ANN model is the best alternative to predict serum digoxin concentrations in PDA.
38	Zhoupeng Ren et al.	Observational study	[52]	CHD	-	-	This study’s use of two machine models reveals a link between CHDs in Beijing and maternal exposure to ambient particulate matter with an aerodynamic diameter of less than 10 m (PM10).
39	Hui Shi et al.	Observational study	[53]	CHD	-	The ML model assists in deciding on specific therapy and nutritional follow-up strategies while making early forecasts of malnutrition in children with CHD at 1 year postoperative.	-
40	Lei Huang et al.	Observational study	[54]	CHD	-	-	In post-Glenn shunt patients with suspected mean pulmonary arterial pressure >15 mmHg, the preoperative cardiac computed tomography (CT)-based RF model exhibits good performance in the prediction of mean pulmonary arterial pressure, potentially reducing the requirement for right heart catheterization.
41	Andreas Hauptmann et al.	Observational study	[55]	CHD	Real-time radial data artefact suppression using a residual U-Net could aid in the widespread use of real-time CMR in clinical settings. Children and sick people who are unable to hold their breath would benefit most from this.	-	-
42	Gerhard-Paul Diller et al.	Observational study	[56]	ACHD	-	-	Machine learning algorithms that have been trained on big datasets can be useful for estimating prognosis and possibly directing therapy in ACHD.
43	Weize Xu et al.	Observational study	[57]	CHD	The precise classification of CHD is completed using a heart sound segmentation method based on PCG segment to achieve the segmentation of cardiac cycles. The accuracy, sensitivity, specificity, and f1-score of classification for CHD are, respectively, 0.953, 0.946, 0.961, and 0.953, which demonstrate that the suggested technique performs competitively.	-	-
44	Daniel Ruiz-Fernández et al.	Observational study	[58]	Pediatric cardiac surgery	-	Future difficulties, or even death, could be prevented with the use of AI-based decision support algorithms when classifying the risk of congenital heart surgery.	-
45	Sukrit Narula et al.	Observational study	[59]	HOCM	Using echocardiographic data, machine learning algorithms can help distinguish between physiological and pathological remodelling patterns in hypertrophic cardiomyopathy (HCM) and physiological hypertrophy seen in athletes (ATH).	-	-
46	Sumeet Gandhi et al.	Review	[60]	Cardiology	Automation has been introduced into many vendor software systems to increase the precision and effectiveness of human echocardiogram tracings.	-	-
47	Kipp W Johnson et al.	Review	[61]	Cardiology	Because doctors will be able to analyze a greater volume of data in greater depth than ever before, AI will result in better patient care. Physicians will benefit from the streamlined clinical treatment provided by reinforcement learning algorithms. Unsupervised learning developments will allow for a far more thorough definition of patients’ problems, which will ultimately result in a better choice of treatments and better results.	-	-
48	Peter Kokol et al.	Review	[62]	Pediatric developmental disorders, oncology, emergencies	The use of AI in pediatrics led to better clinical outcomes, more precise and swifter diagnoses, better decision making, and more sensitive and specific identification of high-risk patients.	-	-
49	Chen Chen et al.	Review	[63]	General	Different cardiac anatomical features, such as the heart ventricle, atria, and vessels, can be segmented using deep learning algorithms that are applied in three main imaging modalities: MRI, CT, and ultrasound.	-	-
50	Chang AC et al.	Editorial	[64]	Pediatric heart diseases	The subspecialty that will gain the most from future technologies and AI approaches is pediatric cardiology, hands down.	-	-
51	Diller GP et al.	Observational study	[65]	TOF	-	Automated evaluation of cardiac magnetic resonance (CMR) imaging parameters using machine learning techniques based in two dimensions to predict prognosis in TOF.	In patients with corrected tetralogy of Fallot, automated analysis using machine learning algorithms may replace labor-intensively obtained imaging parameters from cardiac magnetic resonance (CMR) (ToF).
52	Eynde J et al.	Perspective article	[12]	CHD	-	When AI is combined with mechanistic models to describe complicated interactions among variables, medically based data can be utilized to identify trends and predict late problems such arrhythmias and congestive heart failure as well as survival.	
53	Zhang et al.	Observational study	[66]	TOF	-	-	The patch size, shape, and position optimization technique used in pulmonary artery-enlarging repair surgery using generative adversarial networks (GANs) is more accurate and produces superior clinical results.
54	Asmare MH et al.	Observational study	[67]	RHD	The automatic auscultation and categorization of the heart sound as being normal or rheumatic is performed using a deep learning method based on convolutional neural networks. It is not necessary to extract the first, second, or systolic and diastolic heart sounds when classifying un-segmented data.	-	-
55	Lakhe A et al.	Observational study	[68]	CHD	An adaptive line enhancement approach is used by a digital stethoscope to digitally amplify, record, examine, play back, and process heart sounds.	-	-
56	A. Arafati	Review	[69]	CHD	AI-based methods for analyzing cardiac MRI data have the potential to be very effective and error-free.	-	-
57	Pyles Lee at al	Observational study	[70]	CHD	The viability of using the cloud-based HeartLink system to distinguish between pathologic murmurs caused by CHD and typical functional cardiac murmurs was demonstrated in the proof-of-concept study.	-	-
58	Andrisevic N et al.	RCT	[71]	CHD	With a specificity of 70.5% and a sensitivity of 64.7%, an AI-based diagnostic system can distinguish between healthy, normal heart sounds and abnormal heart sounds.	-	-
59	M El-Segaier et al.	RCT	[72]	CHD	First and second heart sounds are detected by an AI algorithm. As benchmarks for detection, R- and T-waves were used.	-	-
60	Sukryool Kang et al.	Observational study	[73]	CHD	With 84–93% sensitivity and 91–99% specificity, the discussed AI algorithm correctly diagnosed Still’s murmur using the jackknife approach based on 87 Still’s murmurs and 170 non-murmurs.	-	-
61	Patricia Garcia-Canadilla et al.	Observational study	[74]	CHD	By enhancing picture capture, quantification, and segmentation, ML methods can enhance the evaluation of fetal cardiac function and help with the early detection of fetal cardiac anomalies and remodelling.	-	-
62	Hong S et al.	Review	[75]	General	ECG tasks including disease diagnosis, localization, sleep staging, biometric human identification, and denoising have all been tackled using deep learning systems.	-	-
63	Bodenhofer U et al.	Observational study	[76]	CHD	Machine learning technologies can more accurately predict the results of valve surgery treatments.	-	Machine learning technologies can more accurately predict the results of valve surgery treatments.
64	Sravani Gampala et al.	Review	[77]	CHD	AI may be useful to radiologists, but it will not replace them.	-	-
65	J. van den Eynde et al.	Perspective	[78]	CHD	Medicine-based evidence has the potential to transform medical decision making.	-	-
66	Mingming Ma et al.	Observational study	[79]	Dilated Obstructed Right Ventricle	Using intelligent navigation technology to STIC volume datasets, FINE can produce and show three unique aberrant fetal echocardiogram images with display rates of 84.0%, 76.0%, and 84.0%, respectively, and therefore may be utilised for screening and remote consultation of fetal DORV.	-	-
67	Zeng X et al.	Review	[80]	CHD	-	-	For effectively predicting problems during pediatric congenital heart surgery, the machine-learning-based model incorporates patient demographics, surgical factors, and intraoperative blood pressure data.
68	Lo Muzio FP et al.	Observational study	[81]	CHD	-	-	AI algorithms can assist surgeons in making decisions during open-chest surgery.
69	Simona Aufiero et al.		[82]	Congenital long QT syndrome	DL models have the potential to help cardiologists diagnose LQTS.	-	-
70	Dias RD et al.	Review	[83]	Cardiology	Machine learning will be used in high-tech operating rooms to improve intra-operative and post-operative outcomes.	-	-
71	Wang T et al.	Observational study	[84]	Kawasaki Disease	A machine learning-based model based on patient data predicts intravenous immunoglobulin resistance in Kawasaki disease.	-	-
72	João Francisco B S Martins et al.	Observational study	[85]	RHD	When the advantage of a 3D convolutional neural network was compared with the benefit of 2D convolutional neural network, the accuracy was 72.77%.	-	-
73	Ghosh P et al.	Review	[86]	MIS-C and Kawasaki disease	Targetable cytokine pathways revealed by the ViP signatures in MIS-C and Kawasaki pinpoint crucial clinical (reduced cardiac function) and laboratory (thrombocytopenia and eosinopenia) indicators to assist monitor severity.	-	-

**Table 2 jcm-11-07072-t002:** AI algorithms in Pediatric Cardiology.

Serial No.	Authors	Reference	AI Algorithms	Algorithm Functions	Pediatric Pathology Assessed
1	Rima Arnaout et. al	[68]	DL Classifier	A deep learning classifier model predicting probable diagnostic outcomes based on real-time imaging or retrospective data	Congenital heart diseases
2	Mori H, Inai K, et. al.	[35]	Convolutional Neural Networks (CNN) & Long Short-term Memory Models (LSTM)	ECG data utilized by CNN to extract waveform shapes that are further classified by LSTM to find ECG features predicting pathology	Atrial septal defect
3	Zuercher M, Ufkes S, et.al.	[69]	Echo-Net Dynamic Model	Using an echocardiogram databank with other cardiac parameter data, the model predicts left ventricular ejection fractions.	LVEF defects in dilated cardiomyopathies
4	Sepehri AA, Hancq J, et. al.	[70]	Arash-Band Method	Specific frequency bands, Arash bands, are used to analyze heart sound energy from pathological murmurs to predict CHDs.	Congenital heart diseases
5	Wang SH, Wu K, Chu T, et al.	[29]	Structurally optimized Stochastic pooling convolutional neural network	Cardiac magnetic imaging data are classified based on a trained convolutional neural network that allows TOF diagnosis.	Tetralogy of Fallot
6	Ko WY, Siontis KC, Attia ZI, et al.	[71]	Convolutional Neural Network enabled ECG	Utilizing 12-lead ECG data to train a convolutional neural network resulting in a model that ascertains HCM diagnosis	Hypertrophic cardiomyopathy
7	DeGroff CG, Bhatikar S, et al.	[72]	Artificial Neural Network (ANN)	Auscultatory data fed into a trained artificial neural network allows classification of normal vs. pathological heart sounds	Pediatric heart murmurs
8	Na JY, Kim D, Kwon AM, et al.	[73]	Light Gradient-Boosting machine (L-GBM)	A decision tree-based algorithm that utilizes prior weaker models to classify data and predict a diagnosis	Patent ductus arteriosus
9	Sepehri AA, Gharehbaghi A, et. al.	[74]	Multi-layer Perceptron (MLP) Neural Network Classifier	An artificial neural network that processes input data through hidden layers to extract and sort data leading to precise segmentation of heart sounds	Pediatric heart sounds
10	Chou, FS., Ghimire, L.V.	[75]	Random Forest Algorithm	A supervised ML algorithm that uses decision trees that are trained using a combination of learning models to aid in precision diagnostic indicators	Pediatric myocarditis
11	Ali F, Hasan B, Ahmad H, et al.	[76]	Long short-term memory (LSTM) recurrent neural network	A recurrent neural network which is trained to retain and utilize past input with concurrent data to recognize patterns for diagnostic predictions	Pediatric rheumatic heart disease

## Data Availability

All data available within this manuscript.

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
