# Peer review of "Artificial Intelligence in Pediatric Cardiology: A Scoping Review"

_jcm, 2022, doi:10.3390/jcm11237072_

Round 1

Reviewer 1 Report

Artificial intelligence in pediatric cardiology is still new concept for most pediatric cardiologists and cardiac surgeons. The content of this manuscript definitely will deliver detailed concept to the readers of this journal. 

1.     English should be checked again because there are many mis-spellings in the manuscript.

2.     Abbreviations should be checked again. CHD and AI were most frequently used abbreviation, but which were not consistent throughout the manuscript.

3.     Table 1 is too redundant. I’m not sure that all the references should be addressed in the table 1. For better readability, the content of Table 1 is better to be concise.

4.     In the 182nd line, ECHO is not correct abbreviation. Please correct as full spelling.

5.     In the 263rd line, ‘in cardiac surgery” was duplicated.

6.     In Figure 1, there is incomplete word of ‘Identificati’. Please correct this.

Author Response

Comment 1. English should be checked again because there are many mis-spellings in the manuscript.

Reply: Thank you for pointing out the error. We have proofread the manuscript in detail again, and to our surprise, we found many such errors which we skipped. We apologize for these errors; we have extensively revised the manuscript to correct them, and we assure you the current version is free of any spelling, grammar, or formatting errors (changes tracked in the revised manuscript).

Comment 2. Abbreviations should be checked again. CHD and AI were most frequently used abbreviation, but which were not consistent throughout the manuscript.

Reply: Thanks for pointing it out; we have made edits to cover it, and each abbreviation used has now been expanded at the point of its first usage. We have also made changes to ensure consistency in the usage of abbreviations allowing better readability (changes tracked in the revised manuscript).

Comment 3. Table 1 is too redundant. I’m not sure that all the references should be addressed in the table 1. For better readability, the content of Table 1 is better to be concise.

Reply: Thanks for the detailed suggestions. We agree with the respected reviewer, but at the same time, as the reviewer addressed themselves, we need such a table in the current study design to condense the compiled data in the best way. The table looks massive but provides a holistically compiled summary of the literature on the topic. We have made changes to hone the content and make it crisp. We have also edited grammar to ensure better readability. We are hopeful that the reviewer and the editor(s) will like it better now. We are confident that the journal's readership will relish such a detailed table with a stratified summary of the literature on the topic in a single place.

Comment 4. In the 182nd line, ECHO is not correct abbreviation. Please correct as full spelling.

Reply: Thank you for pointing it out; we have made the correction as per the suggestion.

Comment 5. In the 263rd line, ‘in cardiac surgery” was duplicated.

Reply: Thank you for pointing it out; we have made the correction as per the suggestion.

Comment 6. In Figure 1, there is incomplete word of ‘Identificati’. Please correct this.

Reply: Thank you for pointing it out. It was hidden during text-box resizing and converting the PRISMA chart as an image. We have now made the correction as per the suggestion (attaching an updated image for the same as well).

Reviewer 2 Report

I would like to congratulate the authors for putting in a novel systematic review assessing the utility of artificial intelligence in paediatric cardiology. However, I believe certain issues could be fixed:

1.The table in the manuscript needs to be revised. There should be more focus on diagnostic accuracy of different studies. I would recommend having multiple tables each focusing on the different utility of AI as highlighted in the text.

2. In the review, I would focus more on different machine learning algorithms. Explaining different ML algorithms with figures would be essential for better understanding.

Author Response

Comment 1: I would like to congratulate the authors for putting in a novel systematic review assessing the utility of artificial intelligence in paediatric cardiology. However, I believe certain issues could be fixed.

Reply: We are glad that the respected reviewer liked our work, and we express our thanks for their valuable suggestions to help improve our manuscript.

Comment 2: The table in the manuscript needs to be revised. There should be more focus on diagnostic accuracy of different studies. I would recommend having multiple tables each focusing on the different utility of AI as highlighted in the text.

Reply: The studies describing the role of AI in pediatric cardiology have been compiled in Table 1 and have been stratified to express the use of AI for diagnosis, prognosis, and planning an intervention in pediatric cardiology. Similarly, Table 2 describes AI algorithms in Pediatric Cardiology.

We attempted to split the table, but it made the content disorganized and failed to provide the holistic picture we aimed for. We, therefore, decided to stay with the current format of the table and made the content crisp. We also rephrased sentences for better readability. We hope the reviewer and/or editor(s) will like it even more now. We are confident that the journal's readers will appreciate such a detailed table containing a stratified summary of the literature on the topic in one place.

Comment 3: In the review, I would focus more on different machine learning algorithms. Explaining different ML algorithms with figures would be essential for better understanding.

Reply: We thank the reviewer for his detailed insights; we made Table 2 highlight the role of AI algorithms, but now as per the suggestions, we have added a section “3.7. AI algorithms in pediatric cardiology” with an illustration “Figure 3”. The previous Figure 3 stands for Figure 4 now. We believe that with this illustration and addition, the review provides a complete comprehensive picture, and we hope that it meets the expectation of the reviewer.

We again express our gratitude to the reviewers and the editor for the detailed help in improving the work. We look forward to hearing from you regarding our submission and responding to any further questions and comments you may have.